# ACHIEVING CERTIFIED ROBUSTNESS AND MAINTAINING CLEAN ACCURACY VIA VANILLA MODEL GUIDE

## ABSTRACT

Certified robustness can provide theoretical defense guarantees for deep neural network models against adversarial examples within a certain perturbation range. However, existing research on obtaining certified robustness requires specialized certified robust training from scratch for DNNs models. This approach significantly decreases the clean accuracy of normal inputs compared to vanilla models trained with vanilla training, affecting the main inference task of DNNs models and causing practical difficulties for security methods. We propose a practical training method that aims to obtain certified robustness while maintaining clean accuracy. This method involves adding a pre-trained vanilla model and applying singular value decomposition (SVD) to the weight matrices of each network layer of the vanilla model. This process yields rotation matrices and singular values that respectively affect clean accuracy and certified robustness. The vanilla model is used as a guide model, establishing a knowledge transfer process based on the similarity of rotation matrices between the guide model and the certification model that obtains certified robustness. In order to select important rotation matrix information and reduce computational cost, a low-rank approximation is used for practical knowledge transfer. Experimental results demonstrate that our approach significantly improves clean accuracy while only slightly reducing certified accuracy.

## 1 INTRODUCTION

Deep learning has achieved great success in many fields, such as computer vision, natural language processing, speech recognition. Increasingly, deep neural network models (DNNs) have been trained and deployed in various software and hardware environments to bring commercial benefits and social services. However, these DNNs models through vanilla training (referred to as vanilla models) are vulnerable to adversarial examples, where small perturbations are added to the clean input to cause the vanilla model to incorrectly output the result (Szegedy et al., 2013). Many heuristic defense methods have been proposed to block adversarial examples, such as gradient masking (Papernot et al., 2017), adversarial training (Madry et al., 2017; Kurakin et al., 2016; Goodfellow et al., 2014), and defensive distillation (Papernot et al., 2016). However, the shields constructed by these methods can be easily broken by adaptive attack methods (Liu et al., 2022; Croce & Hein, 2020; Tramer et al., 2020), which raises security concerns for a large number of deployed vanilla DNN models. In particular, these vanilla models are applied to safety-critical domains such as autonomous driving, medical testing, face recognition.

The existing deep learning security community tries to provide theoretical guarantees for defending against adversarial examples, and certified robustness has been heavily studied for providing invariance in the output results of DNNs model within a certain range of perturbations. The defense methods of certified robustness can be simply divided into two categories: the first category is the determination method of computing the propagation interval of the perturbation in DNNs model (e.g., (Huang et al., 2021; Lee et al., 2020; Tsuzuku et al., 2018; Gowal et al., 2019)), the second category is the probabilistic method using randomized smoothing(e.g., (Salman et al., 2019; Cohen et al., 2019; Lecuyer et al., 2019)). The remaining other methods are less used because of the difficulty of scaling and the high computational effort (e.g., (Wang et al., 2018; Weng et al., 2018; Pulina & Tacchella, 2010)). The first type of method to determine the perturbation interval is to calculate the Lipschitz constant of the DNNs model to measure the change range of the perturbation on the value of the output result. Because the Lipschitz constant of the vanilla model is too large to cause the

adversarial examples, so it is necessary to adjust Lipschitz constant by training the DNNs model from scratch. It causes the clean accuracy of the main inference task because the DNNs model is too regularization. The decrease of clean accuracy makes the DNNs model's practicality seriously affected. The second class of probabilistic methods using randomized smoothing requires adding a smooth classifier to the input before it enters the vanilla model. This certified robust approach is uncertain, i.e., there will be adversarial examples to pass the certified robustness test.

We then propose a practical training method that aims to obtain certified robustness while maintaining clean accuracy. We refer to the models that achieve certified robustness through certified robust training as ***certification model***, and we designate the extensively pre-trained models as ***vanilla model***. (1) We start by adding a pre-trained vanilla model, which has the same network structure as the certification model, as the guide model. Then, we perform singular value decomposition on the weight matrices of each network layer of the vanilla model, obtaining singular values and rotation matrices. By establishing the similarity of rotation matrices between each network layer of the vanilla model and the certification model, we transfer knowledge to guide the clean accuracy of the certification model. (2) To select important rotation matrix information, we only use the low-rank approximation of the rotation matrices from the vanilla model for knowledge transfer. For each network layer of the certification model, we approximate its rotation matrix using the spectral norm computed during certified robust training, reducing the computational cost of the entire process.

Our method offers practical solutions for a wide range of scenarios that require high-performance clean accuracy inference tasks and provide certified robustness in terms of security. Our contributions are summarized as follows.

- We introduce Duet, which employs a pre-trained vanilla model as a guide teacher to maintain the clean accuracy capability of the model requiring certified robustness (referred to as the certification model).

- We perform singular value decomposition on the weight matrices of the network layers in the pre-trained vanilla model to obtain rotation matrices that contribute to clean accuracy.

- In certified robust training, we establish knowledge transfer of rotation matrix similarity between each network layer of the vanilla model and the certification model to maintain the clean accuracy of the certification model.

- The experiment demonstrates that Duet achieves 3.76% improvement in clean accuracy compared to the previously trained certified robustness model that only utilized global Lipschitz regularization for training. It performs on par with models trained using local Lipschitz regularization which the significant computational and memory costs make it difficult to put into practical use. Meanwhile, there is a slight decrease of 0.93% in certified accuracy.

## 2 PRELIMINARY

In this section, we provide the background for the Duet method by introducing the concepts and symbols of deep learning and certified robustness involved in this paper.

**DNNs model architecture.** In this paper, we focus on researching DNNs model architecture for classification tasks. This architecture with $L$ layers can be seen as a mapping function $F(x, W)$ from $\mathbb{R}^{N_1} \Rightarrow \mathbb{R}^{N_2}$, where $x \in X \subset \mathbb{R}^{N_1}$ is the input, $z \in Z \subset \mathbb{R}^{N_2}$ is the output logit vector, and $W$ represents the weight parameters of the DNN model. The maximum value $z_y$ of the output logit vector $z$ is associated with the correct classification label result $y \in Y$, $Y$ represents all relevant predicted labels. The function $F(x, W)$ is defined as,

$$F(x, W) = \phi(\phi(x, W^0)..., W^{l-1}). \tag{1}$$

$W = \{W^{l \in [0, L-1]}\}$ represents the trainable weight parameters from the first to the last layer. $\phi(\cdot) = \max(\cdot, 0)$ is the element-wise ReLU activation function. In this paper, we differentiate the parameters involved in the two types of DNN model architectures: vanilla model, certification model using subscripts "van" and "cer" respectively. The relevant parameter symbols are $F_{van}$, $F_{cer}$, $W_{van}$, $W_{cer}$, $z_{van}$ and $z_{cer}$.

**Lipschitz constant of neural network.** The Lipschitz constant is used to determine the maximum possible change in the output $z$ of the function $F$ when the input $x$ changes. In the field of deep

learning, Lipschitz constant is commonly used to measure the generalization and robustness of neural networks. The Lipschitz constant of function $F$ is,

$$Lip(F, X) := \sup_{x_0, x_1 \in X, x_0 \neq x_1} \frac{||F(x_0) - F(x_1)||}{||x_0 - x_1||}. \tag{2}$$

The Lipschitz constant of the neural network $F$ is calculated by multiplying the spectral norm $||W||^l$ of each linear layer weight matrix $W^l$. The spectral norm $||W||^l$ is usually calculated using the power iteration method or singular value decomposition. For vanilla model and auxiliary model, their Lipschitz constant calculation formulas is,

$$Lip = \prod_{l=0}^{L-1} ||W^l||. \tag{3}$$

**Singular value decomposition.** The matrix A can be explained using Singular Value Decomposition (SVD) as,

$$\begin{bmatrix} & & \\ & W & \\ & & \end{bmatrix} = \overset{U}{\begin{bmatrix} | & | & \\ u_1 & u_2 & \cdots \\ | & | & \end{bmatrix}} \overset{\sum}{\begin{bmatrix} \sigma_1 & 0 & \cdots \\ 0 & \sigma_2 & \cdots \\ \vdots & \vdots & \ddots \end{bmatrix}} \overset{V^T}{\begin{bmatrix} - & v_1 & - \\ - & v_2 & - \\ & \vdots & \end{bmatrix}} \tag{4}$$

SVD is a matrix factorization method that decomposes $A$ into the product of three matrices: $U$, $\sum$, and $V^T$. $U$ and $V$ are orthogonal matrices containing the left and right singular vectors of $A$, respectively, while $\sum$ is a diagonal matrix containing the singular values $\sigma$ of $A$.

**Certified robustness.** To determine whether a DNN model's prediction is certified robustness, we compute the minimum boundary value $M_{F,x}$ under the influence of the adversary. We subtract the output logit vector $z_{y'}$ corresponding to each other prediction label $y'$ from the output logit vector $z_0$ associated with the correct prediction label $y_0$. The formula for computing the latest boundary value $M_{F,x}$ is,

$$M_{F,x} = \min_{x \in Ball_p(x_0, \epsilon)} (z_{y_0} - z_{y'}). \tag{5}$$

Where, $z_{y_0} = F(x, W)_{y_0}$, and $z_{y'} = F(x, W)_{y'}$. If $M(F, x) > 0$ for all $y' \in [Y] \setminus \{y_0\}$, we say that the DNN model $F(x, W)$ is certified robust under the influence of the $(l_p, \epsilon)$-*adversary*. This means that the correct prediction label outputted in inference will not be changed.

**Certified robust training.** For the DNNs model $F_{van}(x, W)$ trained by vanilla training methods, since its minimum margin $M_{F,x}$ is always $< 0$, the vanilla model does not have certified robustness. To obtain certified robustness for DNNs models, we need to use special certified robust training methods. Certified robust training can continuously optimize the minimum margin value $M_{F,x}$ to make it $> 0$ while training the DNNs model from scratch. Since it is an NP-hard problem to accurately solve the value of $M_{F,x}$, we usually optimize the lower bound of $M_{F,x}$.

**Certified robustness using Lipschitz constant.** For a DNN model, the lower bound of $M_{F,x}$ can be computed using the Lipschitz constant of $F(x, W)$ in order to determine whether the input possesses certified robustness. Previous work (Tsuzuku et al., 2018) has obtained a lower bound for $M_{F,x}$ using the Lipschitz constant of $F(x, W)$ as,

$$M_{F,x} \geq (\min_{y' \in [Y] \setminus \{y_0\}} (z_{y_0} - z_{y'}) - \sqrt{2}Lip(F, X)\epsilon). \tag{6}$$

However, using only the Lipschitz constant to calculate the lower bound of $M_{F,x}$ can result in a loose lower bound, which decreases the certified accuracy of DNN models. Subsequent work(Lee et al., 2020) combines Lipschitz constant and interval propagation to solve for a more accurate outer bound of the output logit with perturbed inputs. By using this outer bound, the worst-translated logit, denoted as $z^*$, can be obtained. $z^*$ can then provide a tighter lower bound for $M_{F,x}$. Each term in $z^*$ can be defined as,

$$z_{y'}^* = F(x, W) - \min_{z \in \hat{z}(\mathbb{B}(x))} (z_{y_0} - z_{y'}). \tag{7}$$

## 3 RELATED WORK

LMT (Tsuzuku et al., 2018) determines the maximum change range of perturbation by calculating the Lipschitz constant of DNNs model, and adds the Lipschitz constant to the corresponding logit except for the correct class label during feedback training to enhance certified robustness in certified robust training. GloRo (Leino et al., 2021) calculates the Lipschitz constant of the DNNs model and constructs a new output logit vector for a specific class based on it, which is used for certified robust training to enhance certified robustness. BCP (Lee et al., 2020) suggested that using only the global Lipschitz constant in certified robust training could lead to over-regularization of the DNNs model, resulting in reduced generalization performance. BCP not only uses the global Lipschitz constant but also calculates the corresponding perturbation interval propagation (box constraint bound) for each input using the Interval Bound Propagation (IBP) (Gowal et al., 2019) method. By taking the intersection of the two, BCP can obtain a tighter propagation interval, which further enhances certified robustness. Local Lipschitz Bounds (Huang et al., 2021) suggests that previous work on certified robust training has focused on using globally calculated Lipschitz constants that can be computationally efficient, but such an approach results in over-regularized DNNs models, leading to reduced clean accuracy. To compute local Lipschitz bounds more efficiently, Local Lipschitz Bounds analyzes non-linear functions such as ReLU and linear layer weight matrices, eliminating rows and columns of the weight matrix output constants to obtain a more accurate local Lipschitz constant. From another perspective, for example, the excessive warm up (Shi et al., 2021) process during certified robust training determines a new weight initialization process and adds a Batch Normalization operation for each layer, similarly by changing the conditions of other training based on the computation of the perturbation propagation interval certified robustness. The work also include (Zhang et al., 2022; Mirman et al., 2021; Mao et al., 2023; Feng et al., 2022; Cullen et al., 2022; Li et al., 2022; Zeng et al., 2023; Zhang et al., 2019). The propagation interval method mentioned earlier provides deterministic certified robustness, while another class of methods based on randomized smoothing provides probabilistic certified robustness, meaning that inputs satisfying certified robustness may still be adversarial examples. (Cohen et al., 2019) achieved $l_2$ norm based certified robustness for DNNs models on ImageNet datasets (Deng et al., 2009) by establishing Gaussian noise smoothing, which makes certified robustness for DNNs models scalable. (Lecuyer et al., 2019) established the theoretical connection between certified robustness and differential privacy by treating images as a database and each pixel in the image as a tuple in the database to analyze the impact of perturbation on the image using theoretical analysis. (Carlini et al., 2022) achieves higher clean accuracy for DNNs models by using a pre-trained denoising diffusion probabilistic model for denoised smoothing. Other works to improve the certified robustness of probabilistic methods include (Jeong & Shin, 2020; Lee, 2021; Salman et al., 2020; Zhai et al., 2020; Yang et al., 2020; Jeong et al., 2021).

## 4 METHOD

### 4.1 OVERVIEW

Figure 1 illustrates the entire workflow. It can be divided into two phases. The first phase involves obtaining and loading a pre-trained vanilla model. The second phase consists of adding an certification model to construct an knowledge transfer process and performing certified robust training.

In the first phase (step ①), our method initializes the model structure of the vanilla model and loads the pre-trained parameters of the vanilla model. Then it calculates the singular value decomposition of the weight matrices of the vanilla model.

In the second stage (step ②~③), during certified robust training of the knowledge transfer process, our method first adds an certification model with the same model structure as the vanilla model. Then, it combines the vanilla model and the certification model to form the knowledge transfer process (step ②). To preserve clean accuracy during certified robust training, our method also maintains the similarity of the unit vectors of the SVD decomposition rotation matrices between the vanilla model and the certification model (step ③). The logit vector and Lipschitz constant of the knowledge transfer process are calculated to obtain the worst logit for certified robust training and achieve certified robustness (step ④).

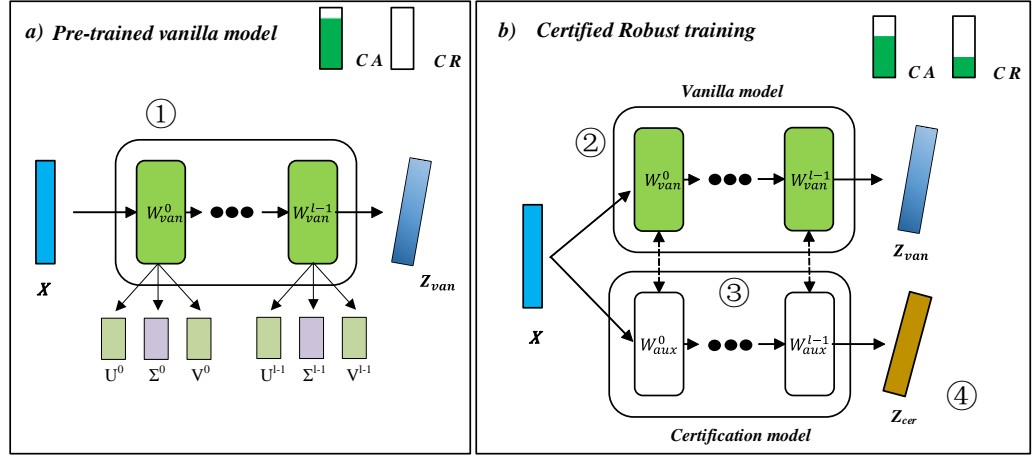

Figure 1: **Overview**

## 4.2 WHY DOES CERTIFIED ROBUSTNESS TRAINING LEAD TO A DECREASE IN CLEAN ACCURACY?

For the certified robust training method based on Lipschitz constant for interval propagation estimation, the Lipschitz constant of the certification model can be determined. Additionally, considering the box constraints of the actual inputs, the worst logit can be obtained for the purpose of backward propagation. However, this allows the network structure of the certification model to minimize the influence of adversarial perturbations on the output margin by scaling the spectral norm, i.e., the maximum singular value of the weight matrices, as the perturbations propagate through each layer of the network. This allows the network structure of the certification model to minimize, as much as possible, the impact of adversarial perturbations on the output margin. It achieves this by scaling the spectral norm, which is the maximum singular value of the weight matrices, as the perturbations propagate through each layer of the network. However, this approach also leads to the scaling range of different directions in the weight matrices becoming similar. As a result, the certification model finds it difficult to rotate the values in a vector to the appropriate positions for varying scales, which affects numerical magnitude changes.

For weight matrices, their ability to accurately adjust the magnitude of input vector changes is not solely determined by the variation in the maximum scale, i.e., singular values. From the perspective of SVD, it is also influenced by the rotation matrix's effect on the input vector in the space. By performing SVD on the weight matrices, we can observe that similar singular values make the certification model less sensitive to the scale changes of each layer's weight matrices. This prevents the upper bound of perturbation propagation from affecting the non-negativity of the margin. However, this also means that a well-trained certification model relies more on the rotation effect of the weight matrices. In other words, it allows the values of the input vector to be correctly scaled after coordinate transformation. However, traditional certified robust training only takes into account the influence of singular values on certified robustness, neglecting the role of rotation matrices. This leads to a decrease in clean accuracy compared to the vanilla model for the certification model.

When certified robust training is completed, existing methods for computing worst logit $z^*_{cer}$ approximate the calculation process using the weight matrix $W_{cer}$, obtained by performing singular value decomposition on $W_{cer}$ as,

$$\begin{bmatrix} & & \\ & W_{cer} & \\ & & \end{bmatrix} = \overset{U}{\begin{bmatrix} | & | & \\ u_1 & u_2 & \cdots \\ | & | & \end{bmatrix}} \overset{\Sigma}{\begin{bmatrix} \sigma_1 & 0 & \cdots \\ 0 & \sigma_1 & \cdots \\ \vdots & \vdots & \ddots \end{bmatrix}} \overset{V^T}{\begin{bmatrix} - & v_1 & - \\ - & v_2 & - \\ & \vdots & \end{bmatrix}} \tag{8}$$

In other words, the maximum singular value $\sigma_1$ is the same as the other singular values, which results in equal variations in the weight matrix $W_{cer}$ across all directions.

### 4.3 LOW-RANK APPROXIMATION OF VANILLA ROTATION MATRICES.

Decomposing the weight matrices of each layer in the vanilla model using SVD can be done in the offline stage, which doesn't require training or inference time. For the weight matrices of the vanilla model, low-rank approximation can be achieved by using a small number of singular values and the orthogonal unit vectors from the rotation matrices for recovery.

In our approach, the rotation matrices of the vanilla model are already capable of appropriately rotating input vectors. Similarly, during computation, it is not necessary to use all orthogonal unit vectors from the rotation matrices. Only a few orthogonal unit vectors are needed to approximate the recovery of rotation matrices $U$ and $V$. The low-rank approximation of the rotation matrices is as,

$$\sum_{i \leq r} u_i v_i^T = UV^T. \tag{9}$$

Low-rank approximation not only selects the important information from the rotation matrices but also reduces the computational and storage costs.

### 4.4 PRESERVATION OF SIMILARITY BETWEEN ROTATION MATRICES.

For the vanilla model, its high performance classification ability for clean input is expressed by the parameter values of the weight matrices of each layer, and the change in the values of these weight matrices brings high performance classification ability in the normal training process, while in the previous certified robust training process, these weight matrices are affected by the Lipschitz In the previous certified robust training process, these weight matrices were affected by Lipschitz constant reinforcement, which made their changes to the input become small, resulting in poor classification results for the clean input. To solve this problem, our method makes the parameters of the weight matrices of each layer of the certification model handle clean input well by making SVD decomposition rotation matirx $U$ and $V^T$ of each layer of the certification model as similar as possible to the same layer weight rotation matrices decomposition the vanilla model, so that our method's retention of clean accuracy This is also to strengthen the role of the weight parameters in the conflict between certified robustness and clean accuracy. The different $U$ and $V_T$ of Rota matrix similarity is,

$$Rota_{van} = U_{van}^l (V_{van}^T)^l x,$$
$$Rota_{cer} = U_{cer}^l (V_{cer}^T)^l x. \tag{10}$$

However, it is not feasible to directly calculate the similarity between the rotation matrics of each layer of the certification model and each layer of the vanilla model. This makes the output of each layer of the vanilla model must be adjusted, otherwise it will make it difficult for the output of each layer of the certification model to learn the appropriate representation and thus reduce our method's performance. The simplified method is based on the calculation of the Lipschitz constant, that is, the multiplication of the spectral parametrization of each linear layer gives the lipshcitz constant, and we can obtain this approximate change relationship by adjusting the relationship between the spectral parametrization of each layer of the vanilla model and the spectral parametrization of each layer of the certification model, which makes The entire clean reserved process can be integrated in the previous process of obtaining certified robustness for the vanilla model, which improves the efficiency of the calculation, and the corresponding loss calculation formula is,

$$Loss_{sim} = \sum_{l=0}^{L-1} || \frac{||\sigma_{cer}^l||}{||\sigma_{van}^l||} Rota_{van} - \frac{||\sigma_{van}^l||}{||\sigma_{cer}^l||} Rota_{cer} ||_2. \tag{11}$$

Finally, our full training procedure is presented in Algorithm 1.

---

**Algorithm 1:** Certified Robust Training

**Input**   :Data $X$,Pre-trained Model $F_{van}$ and $Lip_{van}$,certification Model $F_{cer}$
**Output** :Provable Robustness Model $F_{cer}$

1 **for** $i \leftarrow 0$ **to** *epoch* **do**
2     **for** $j \leftarrow 0$ **to** *batchsize* **do**
3        $F_{cer} \leftarrow X_{batch}$
4        $F_{van} \leftarrow X_{batch}$
5        **for** $l \leftarrow 0$ **to** *Layersize* **do**
6           $U_{van}^l, \sigma_{van}, V_{van}^l = SVD(W_{van}^l)$
7           $\sigma_{cer} = SpecNormCal(W_{cer}^l)$
8           $Lip_{cer} = \sigma_{cer} \cdot Lip_{cer}$
9           $W_{cer}^l = W_{cer}^l / \sigma_{cer}$
10          $Rota_{van} = U_{van}^l (V_{van}^T)^l x$
11          $Rota_{cer} = U_{cer}^l (V_{cer}^T)^l x$
12          $loss_{sim} += || \frac{||\sigma_{cer}^l||}{||\sigma_{van}^l||} Rota_{van} - \frac{||\sigma_{van}^l||}{||\sigma_{cer}^l||} Rota_{cer} ||_2$
13        $z_{cer}^* = Worst\text{-}logit(z_{cer})$
14        $Loss_{total} = loss_{z_{cer}} + loss_{sim}$
15        *Upadte Weight Parameters*
16 **return** $F_{cer}$

---

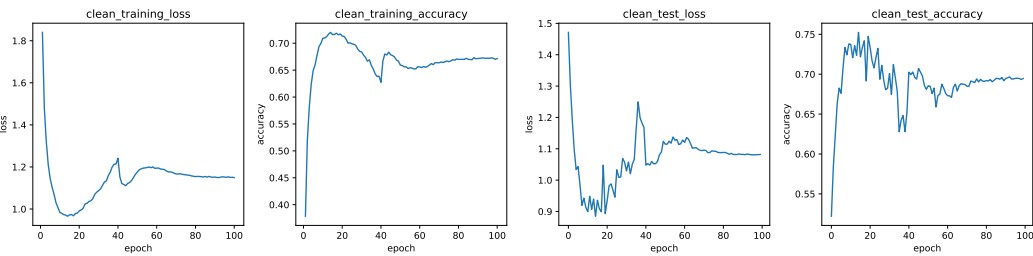

Figure 2: **Duet Clean accuracy**

## 5   EXPERIMENT

In this section, we first introduced the software and hardware environment used, as well as the corresponding information of the DNNs model, including the model, data, algorithm. We compared the certified robust training methods with only global Lipschitz constant(Lee et al., 2020) and local Lipschitz constant(Huang et al., 2021), respectively. In terms of model accuracy, computation time, and memory consumption. We demonstrated that the certified robust training method using vanilla model as guide can effectively improve the clean accuracy of the model while slightly lowering the certified accuracy.

**Experiment setup.** All our experiments were conducted using PyTorch version 2.0.0 on a single NVIDIA A40 GPU. The dataset used was CIFAR-10. Both the vanilla model and the model requiring certified robustness had the same architecture as previous work, which is six Convolutional Layers and two Fully Connected Layers. Depending on the previous work, either ReLU or MaxMin was used as the activation function. The pre-trained vanilla model achieved an accuracy of 90.06%. The $l_2$-norm-based perturbation radius used is $36/255$.

**Clean accuracy.** As shown in Figure 2, in the process of certified robust training, the phenomenon where the clean loss initially increases and then decreases is caused by the larger rotation angles and values of the rotation matrices and singular values corresponding to each layer's SVD decomposition in the vanilla model. In order to make the vanilla model similar to the certified robustness model in terms of rotation matrices, Duet continually adjusts the rotation angles of the weight matrix with respect to the vectors. This adaptation process is necessary for randomly initialized weight matrices.

| Method | Similarity(%) | Lipschitz Constant |
|---|---|---|
| LMT(Tsuzuku et al., 2018) | 51.6 | 19.9 |
| BCP(Lee et al., 2020) | 70.3 | 5.5 |
| LocalBCP(Huang et al., 2021) | 77.8 | 3.8 |
| Our work | 95.6 | 7.8 |

Table 1: Rotation matrix similarity.

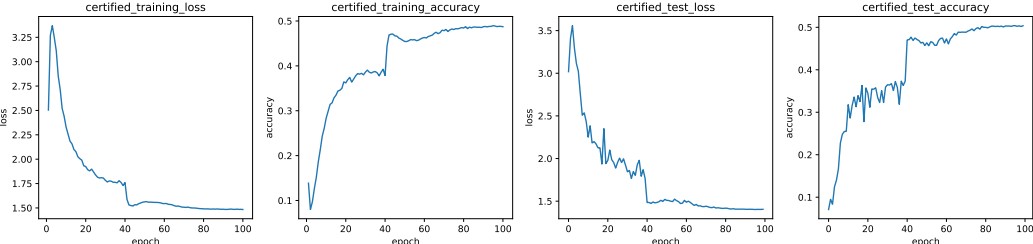

Figure 3: **Duet Certified robustness**

As for clean accuracy, the instability of the similarity of rotation matrices also introduces more numerical fluctuations in the testing process.

**Rotation matrix similarity.** During the previous certified robust training process, the average similarity between the rotation matrices of the certification model and the original vanilla model was 77.8%. However, for our method, the trained rotation matrices achieved a similarity of 95.6%. A higher similarity matrix indicates that the certification model has better grasp of the optimal rotation angles for the input content. Even though the spectral norms of weight matrices at each layer may stretch the output values along each axis, the correct rotation direction still maintains the classification ability for clean accuracy.

**Certified robustness.** As shown in Figure 3, during the certified robust training process, it is observed that both the certified robustness loss and accuracy stabilize for a certain period of epochs. This phenomenon can be attributed to the logit mapping used in the model, which tends to make the outputs of both loss and accuracy converge to some extent. To break this balance, the continuous preservation of similarity in unit vectors within the rotation matrix is crucial. This indicates that the method employed to maintain clean accuracy helps the certified accuracy and loss to break out of saddle points, leading to improved training performance.

**Effective comparison.**

As shown in Table 2, our work has shown a consistent improvement in both certified robustness and clean accuracy compared to other existing methods. Specifically, in terms of certified robustness, compared to the best existing work in certified robustness, our method only incurs a slight decrease of less than 1% in certified accuracy when retaining the auxiliary model's similarity to the rotation matrix unit vectors. Additionally, since the Local Lipschitz Bound calculates the Lipschitz constant for each input sample locally, our method may seem to exhibit a significant decrease. However, in reality, our method directly calculates the global Lipschitz constant and can still be extended to incorporate this approach.

**Computational cost.**

As shown in Table 3, during the training process, the computational cost associated with the similarity calculation of the rotation matrix that we introduced is moderate. In contrast, the highest-certified accuracy method shown in the table, Local Lipschitz Bound, exhibits a rapid increase in computational cost and memory usage as the depth of the neural network increases. For the different network architectures of 4C2F and 6C2F, the corresponding computation times and memory usage are 45.8s and 67.8s, and 21GB and 40.3GB, respectively. This poses challenges to the scalability and practicality of the Local Lipschitz Bound method. In contrast, our work, as well as other existing methods, does not involve the calculation of Lipschitz constants for each input sample at every layer.

| Method | Clean(%) | PGD(%) | Certified(%) |
|---|---|---|---|
| LMT(Tsuzuku et al., 2018) | 63.1 | 58.3 | 38.1 |
| BCP (Lee et al., 2020) (Baseline) | 65.7 | 60.8 | 51.3 |
| LocalBCP(Huang et al., 2021) | 70.7 | 64.8 | 54.3 |
| Our work | 69.46 (+3.76) | 61.8 | 50.37 (-0.93) |

Table 2: Comparison to other certified training algorithms.

| Method | Time(Sec) | Memory(GB) |
|---|---|---|
| LMT(Tsuzuku et al., 2018) | 569 | 8 |
| BCP(Lee et al., 2020) | 14.8 | 3.2 |
| LocalBCP(Huang et al., 2021) | 67.8 | 40.3 |
| Our work | 38.5 | 5.1 |

Table 3: Computation cost.

The computation time and memory cost are not sensitive to the depth of the network in our approach, which enhances its scalability and applicability.

## 6 CONCLUSION

In this paper, We propose a practical training method that aims to obtain certified robustness while maintaining clean accuracy. This method involves adding a pre-trained vanilla model and applying singular value decomposition (SVD) to the weight matrices of each network layer of the vanilla model. This process yields rotation matrices and singular values that respectively affect clean accuracy and certified robustness. The vanilla model is then used as a guide model, establishing a knowledge transfer process based on the similarity of rotation matrices between the guide model and the certification model that obtains certified robustness. In order to select important rotation matrix information and reduce computational cost, a low-rank approximation is used for practical knowledge transfer. Additionally we hope that our method can offers practical solutions for a wide range of scenarios that require high-performance clean accuracy inference tasks and provide certified robustness in terms of security.

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
