# OpenReview forum: "Achieving Certified Robustness and Maintaining Clean Accuracy via Vanilla Model Guide"
_ICLR.cc/2024/Conference — Submitted to ICLR 2024_

### Official Review · Reviewer_wK2j · 2023-10-19

**Soundness:** 2 fair
**Presentation:** 1 poor
**Contribution:** 2 fair
**Rating:** 3
**Confidence:** 3

**Summary:**

This paper proposes a training method for DNNs in order to achieve certified robustness and yet maintain clean accuracy. It uses a "vanilla" model as a guide. The proposal is supported by experimental results.

**Strengths:**

The proposed technique demonstrates an improvement in clean accuracy against the chosen baseline, though at the cost of a small reduction in certified accuracy.

**Weaknesses:**

This paper has a lot of presentation issues - both from poor grammar but also poor presentation of ideas/concepts. Some specific examples are identified below. As such, it makes it very hard to identify what exactly has been done.

That said, even if well presented, the proposed technique results in only a modest increase in certified robustness, and at the expense of clean accuracy drop. Also, it is beaten by techniques like LocalBCP - which admittedly needs more time and memory, but then again so does the proposed technique vs the baseline BCP method.

This arguable improvement on the baselines and the poor presentation lead to my current score. Though happy to revise based on any feedback/updates.

Miscellaneous comments:

LocalBCP is used as a baseline in the tables and yet not mentioned in the text (e.g. under Related Work)

Some comparison with Randomised Smoothing techniques in the experiments might have been good to see.

- Grammatical/presentation issues throughout, for example:
  + "It causes the clean accuracy of the main inference task because the DNNs model is too regularisation"
  +  Section 3 describing related work could be broken into more readable paragraphws
  + The 3rd paragraph of section 4.1 is very unclear
  + 4th and 5th lines of section 4.4 are repeated?
  + "matirx"
  + "Upadte"
  + "..., which make The entrire clean reserved process can be integrated ..."
- "randomized smoothing requires adding a smooth classifier to the input". No, it adds noise to the input
- It is claimed that randomized smoothing is inadequate because it is probabilistic. However, this threshold can be set as per required risk appetite. This is not as big an issue as I think the authors make out
- Issues with presentation of maths:
  + "The matrix A" mentioned above equation (4) has not been mentioned before/defined
  + Typo z_0 should be z_{y_0} above equation (5)
  + \hat z in the minimisation in equation (7) has not been defined
  + Line 8 of Alg 1: Lip_{cer} has not been initialised so this line does not work the first time it is encountered
  + Line 11 of Alg 1: U_{cer} and V_{cer} have not been computed and yet they are used here
  + Line 13 of Alg 1: z^*_{cer} is computed here and yet not used??

**Questions:**

It is stated that for DNNs trained by vanilla methods M_{F,x} is always <0. Doesn't this depend on the parameter \epsilon? ie if epsilon is small enough, it should be able to be certified

There seems to be a misunderstanding of certified robustness. e.g. it is stated ""inputs satisfying certified robustness may still be adversarial examples". However, surely certifying robustness means asserting that if the classifier gives label y at x then it will give label y at all points within a distance epsilon of x. It is nothing to do with x being an adversarial example. Certification is a guarantee that there is no adversarial example within distance epsilon of x.

Why is clean test accuracy better than clean training accuracy in Figure 2. Naively training should be better?

---

### Official Review · Reviewer_KP43 · 2023-10-28

**Soundness:** 2 fair
**Presentation:** 1 poor
**Contribution:** 3 good
**Rating:** 3
**Confidence:** 3

**Summary:**

The paper proposes a novel training approach, termed Duet, that aims to achieve certified robustness for DNNs while retaining clean accuracy. The method leverages a pre-trained vanilla model as a guide for knowledge transfer. By employing Singular Value Decomposition (SVD) on the weight matrices of the vanilla model's network layers, it obtains rotation matrices which, through a low-rank approximation, guide the clean accuracy of a certification model. The results demonstrate a notable improvement in clean accuracy with only a slight trade-off in certified accuracy.

**Strengths:**

- **Novel idea**: the use of a pre-trained vanilla model as a guide is novel and can help bridge the gap between standard/adversarial training and certified training.

**Weaknesses:**

- **Unconvincing experimental section**: The experimental validation lacks depth, with missing datasets and other baseline comparisons (IBP, CROWN-IBP, COAP). In its current form, I am not convinced that the method proposed improves on exististing certified training techniques.
- **Missing related work**: Several works have been studying the limitations of certified training and the resulting gap in standard and robust accuracy, see for example [1],[2],[3]. I think these should at least be mentioned in the introduction, where the accuracy gap is currently not well motivated.
- **Poor presentation**: The paper has typos and grammatical errors, affecting its overall readability. E.g. "It causes the clean accuracy of the main inference task because the DNNs model is too regularization".

[1]  S. Lee, W. Lee, J. Park, and J. Lee. Towards better understanding of training certifiably robust models against adversarial examples. In Advances in Neural Information Processing Systems, 2021.
[2]  Piersilvio De Bartolomeis, Jacob Clarysse, Amartya Sanyal, and Fanny Yang. How robust accuracy suffers from certified training with convex relaxations. Arxiv, 2023.
[3] N. Jovanovi ́c, M. Balunovic, M. Baader, and M. Vechev. On the paradox of certified training. Transactions
on Machine Learning Research, 2022.

**Questions:**

- How does the method compare to interval-based propagation certified training?
- Why is the experimental evaluation restricted to CIFAR-10? How does the method perform on MNIST and TinyImagenet?

---

### Official Review · Reviewer_XSXh · 2023-11-01

**Soundness:** 2 fair
**Presentation:** 2 fair
**Contribution:** 2 fair
**Rating:** 3
**Confidence:** 3

**Summary:**

This paper aims to improve the clean accuracy in L2 deterministic certified robust training. It introduces a pre-trained clean model and applies SVD to weight matrices to yield rotation matrices and singular values that respectively affect clean accuracy and certified robustness. Then a knowledge transfer is performed between the clean model and the robust model to be trained. A low-rank approximation is applied on the rotation matrices for efficiency, and the training aims to preserve the similarity between the rotation matrices. Lipschitz constants are computed for robust training.

**Strengths:**

* The paper presents an interesting understanding on using rotation matrices and singular values for accuracy and certified robustness, respectively.
* With the new training scheme, the clean accuracy is improved, with a small decrease on the certified robust accuracy, compared to a baseline considered in this paper.

**Weaknesses:**

* Experiments are limited. There is only a single dataset (CIFAR-10) with a single epsilon. There are also only several relatively old baselines. See Hu et al., 2023 for more comprehensive experiments and many more recent baselines.
* There lack baselines that also aim to improve clean accuracy in certified robust training. For example, a simple baseline may combine the objective of standard training and certified training. It is unclear if the method proposed in this paper can outperform simple baselines which may also improve the clean accuracy while sacrificing the robust accuracy.
* Section 4.2 ("WHY DOES CERTIFIED ROBUSTNESS TRAINING LEAD TO A DECREASE IN CLEAN ACCURACY?") is not supported by any experiment or theory.

Hu, K., Zou, A., Wang, Z., Leino, K., & Fredrikson, M. (2023). Scaling in Depth: Unlocking Robustness Certification on ImageNet. arXiv preprint arXiv:2301.12549.

**Questions:**

* How can the claims in Section 4.2 be empirically verified?

---

### Official Review · Reviewer_koaj · 2023-11-07

**Soundness:** 2 fair
**Presentation:** 1 poor
**Contribution:** 2 fair
**Rating:** 3
**Confidence:** 4

**Summary:**

This paper aims to improve the clean accuracy of an L2-norm certified defense model, by using SVD decomposition on model weights. The idea is to make the rotation matrices U and V in SVD more similar to those in a naturally trained, non-certified defense model. A loss function is designed to encourage this similarity. In the main algorithm, two pre-trained models, one trained naturally and one trained using certified defense, were given and the loss function finetunes the certified defense model and changed its rotation matrix in SVD to make it more similar to the naturally trained model. The hope is to yield higher clean accuracy with little sacrifice in certified accuracy. Positive results were demonstrated on a single CIFAR-10 model with three baseline methods.

**Strengths:**

1. The paper studies an important topic of improving the clean accuracy for certified defense models. The tradeoff between clean accuracy and certified accuracy is an open question, and this work contributes to this important direction.

2. The insight that the rotation matrix in SVD can be changed while keeping certification guarantees is novel. It could yield a good paper if the idea is nicely explained and executed.

3. The statistics shown in Table 1 about rotation matrix similarity between naturally trained models and certified defense models are informational.

**Weaknesses:**

1. Experiments were done in only a very limited number of models on CIFAR10 with very few baselines (only LMT, BCP and LocalBCP). It is not convincing. You can find more baselines mentioned in this paper [1]. The core results in Table 2 and 3 are from a single model only.

2. Writing needs improvement. I can hardly understand many sentences, and it looks like the draft is very rough. For example, in section 4.4 the sentence “in the previous certified robust training process, these weight matrices are affected by the Lipschitz” is repeated twice. At the end of page 6 there is a long sentence that spans 7 lines with grammatical errors. Algorithm 1 is written in a very informal way. All the factors make the paper hard to understand and follow.

3. Existing work such as GloroNet [2] already discussed a tradeoff between clean accuracy and certified accuracy (such as the TRADES Loss discussed in [2]). A comparison to these simple approaches is missing.

**Questions:**

1. In algorithm 1 it is unclear how you get $U_{cert}$ and $V_{cert}$. Do you actually run SVD? Do you differentiate through the SVD operation?

2. It seems the paper targets L2-norm-based certified defense only. It would be better to mention it early.

3. How to apply this approach to 1-Lipschitz layers such as Cayley [3] or SLL [4]?

4. Can you give more details on how convolutional networks are handled?


References:

[1] Hu, Kai, et al. "Scaling in Depth: Unlocking Robustness Certification on ImageNet." arXiv preprint arXiv:2301.12549 (2023).

[2] Leino, Klas, Zifan Wang, and Matt Fredrikson. "Globally-robust neural networks." International Conference on Machine Learning. PMLR, 2021.

[3] Asher Trockman and J Zico Kolter. Orthogonalizing convolutional layers with the cayley transform. In ICLR, 2021.

[4] Alexandre Araujo, Aaron J Havens, Blaise Delattre, Alexandre Allauzen, and Bin Hu. A unified algebraic perspective on lipschitz neural networks. In ICLR, 2023.

---

### Meta-Review · Area_Chair_KzEv · 2023-12-15

**Metareview:**

The paper proposes a new algorithm to improve the clean accuracy of certified robust models. All reviewers think this is an important question, and the ideas proposed by the paper seems novel. However, all reviewers think the empirical evaluations in this paper is insufficient (lacking baselines and datasets) and the paper is poorly written making it hard to follow. Therefore all reviewers vote for rejection.

**Justification For Why Not Higher Score:**

Insufficient experiments; writing needs to be improved.

**Justification For Why Not Lower Score:**

N/A

---

### Decision · Program_Chairs · 2024-01-16

Reject